# The structural basis for the phospholipid remodeling by lysophosphatidylcholine acyltransferase 3

Qing Zhang[1,7], Deqiang Yao[2,3,7], Bing Rao[2,4,7], Liyan Jian[2,4], Yang Chen[2], Kexin Hu[2], Ying Xia[2,4], Shaobai Li[2], Yafeng Shen[2], An Qin[4], Jie Zhao[4], Lu Zhou[5], Ming Lei [2], Xian-Cheng Jiang[6] & Yu Cao [2,4✉]

As the major component of cell membranes, phosphatidylcholine (PC) is synthesized de novo in the Kennedy pathway and then undergoes extensive deacylation-reacylation remodeling via Lands' cycle. The re-acylation is catalyzed by lysophosphatidylcholine acyltransferase (LPCAT) and among the four LPCAT members in human, the LPCAT3 preferentially introduces polyunsaturated acyl onto the sn-2 position of lysophosphatidylcholine, thereby modulating the membrane fluidity and membrane protein functions therein. Combining the x-ray crystallography and the cryo-electron microscopy, we determined the structures of LPCAT3 in apo-, acyl donor-bound, and acyl receptor-bound states. A reaction chamber was revealed in the LPCAT3 structure where the lysophosphatidylcholine and arachidonoyl-CoA were positioned in two tunnels connected near to the catalytic center. A side pocket was found expanding the tunnel for the arachidonoyl CoA and holding the main body of arachidonoyl. The structural and functional analysis provides the basis for the re-acylation of lysophosphatidylcholine and the substrate preference during the reactions.

[1] CAS Center for Excellence in Molecular Cell Science, Shanghai Institute of Biochemistry and Cell Biology, Chinese Academy of Sciences, University of Chinese Academy of Sciences, 333 Haike Road, Shanghai 201210, China. [2] Institute of Precision Medicine, the Ninth People's Hospital, Shanghai Jiao Tong University School of Medicine, 115 Jinzun Road, Shanghai 200125, China. [3] State Key Laboratory of Oncogenes and Related Genes, Ren Ji Hospital, Shanghai Jiao Tong University School of Medicine, Shanghai 200127, China. [4] Department of Orthopaedics, Shanghai Key Laboratory of Orthopaedic Implant, Shanghai Ninth People's Hospital, Shanghai Jiao Tong University School of Medicine, Shanghai 200011, China. [5] Department of Medicinal Chemistry, School of Pharmacy, Fudan University, Shanghai 201203, China. [6] Department of Cell Biology, State University of New York Downstate Health Sciences University, Brooklyn, NY, USA. [7] These authors contributed equally: Qing Zhang, Deqiang Yao, Bing Rao. ✉email: yu.cao@shsmu.edu.cn

Phospholipids (PLs) make up the major lipids in cellular membranes and in the circulation[1–3]. PLs are synthesized de novo by the Kennedy pathway[4]. The asymmetrical distribution of fatty acids at the *sn-1* and *sn-2* positions is maintained in part by a deacylation-reacylation process first proposed by Lands 60 years ago (Lands' Cycle or PL remodeling)[5,6]. The reacylation is catalyzed by lysophospholipid acyltransferase[5,6]. So far, no structure has been resolved in this set of enzymes. Phosphatidylcholine (PC) is a major PL and its composition in mammalian cell membrane exhibits considerable structural diversity[7,8]. Four lysophatidylcholine acyltransferases (LPCATs), LPCAT1–4, participate PC remodeling[9–13]. Among them, LPCAT3 is the main isoform in major metabolic tissues, including the liver, small intestine, skeletal muscle, macrophage, and adipocyte[12,14–20]. LPCAT3 is regulated by liver X receptor[21], a well-known factor in control of lipogenesis[22]. LPCAT3 activity influences the biology and pathology of these tissues and plays an important role in lipoprotein production in the liver and small intestine[23,24]. Intestinal LPCAT3 activity is required for a gut-brain feedback loop that couples absorption to food intake[25]. LPCAT3 deficiency reduces lipid absorption, thus reducing atherosclerogenic lipoproteins in the circulation[15,25,26]. LPCAT3 deficiency increases insulin sensitivity in skeletal muscle, through modulating plasma membrane PC composition[27], and impairs preadipocyte adipogenesis through activating Wnt/beta-catenin pathway[20]. In terms of atherogenicity, the reported studies are controversy. On the one hand, the expression of LPCAT3 is associated with atherosclerosis progression[28]. On the other hand, although LPCAT3 deficiency in macrophage increases the production of inflammatory and atherogenic cytokines/chemokines, however, this effect may[29] or may not[30] promote the development of atherosclerosis. Collectively, LPCAT3 is a potential target for the treatment of metabolic disorders such as hyperlipidemia and atherosclerosis[23,31]. However, some concerns are noted, for instance, the disruption of LPCAT3 could promote the tumorigenesis in the intestine, probably by increasing the cholesterol biosynthesis in the intestinal stem cells and thus enhancing its proliferation[32]; blocking LPCAT3 might cause fat accumulation in tissues[16].

LPCAT1 and 2 are type II single-pass membrane proteins containing LPA acyltransferase motifs, and LPCAT3 and 4 are multi-pass membrane proteins[23]. LPCAT3 belongs to the membrane-bound O-acyltransferase (MBOAT) family[33], which includes over 10,000 multi-pass membrane proteins found in species ranging from bacteria to mammals (details available at http://pfam.xfam.org/family/MBOAT). LPCAT3, *aka* MBOAT5, is one of the eleven MOBATs found in the human genome and share about 10–20% protein identity with its human relatives. SOAT1 and DGAT1 are two human MBOATs with structures recently solved[34–37], but the evolutionary diversity made it difficult to understand the lysoPLs acyltransferases mechanism using currently available structures as models. To probe the molecular mechanism underlying acyl transfer process in phospholipids remodeling and the unsaturation preference on the acyl to be transferred, we determined the crystal and cryo-EM structure for chicken LPCAT3 and explored the function of the critical residues implied by the structures.

## Results

### The crystal structure of chicken LPCAT3.
The human and chicken LPCAT3 share a protein sequence identity of about 69% (Supplementary Fig. 1). Both chicken LPCAT3 (cLPCAT3) and human LPCAT3 (hLPCAT3) purified in various detergents display significant acyltransferase activities in thin layer chromatography (TLC) assay using arachidonoyl CoA (araCoA) and NBD-labeled 12:0 LPC as substrates where the formation of NBD-PC by acyl transfer were monitored through the relative mobility shift of the sample bands with the fluorescence from NBD group (Fig. 1a and Supplementary Fig. 2a). Compared with hLPCAT3, cLPCAT3 showed better homogeneity in the gel filtration and higher acyltransferase activity in TLC assay, implying superior stability for cLPCAT3. To probe the catalytic mechanism, extensive efforts in protein engineering and crystallization were conducted on hLPCAT3 and cLPCAT3, but most of them failed. When purified in undecyl maltoside (UM) and treated with trypsin digestion, cLPCAT3 was trimmed into a slightly smaller protein core with only two residues at C-terminal removed along with the affinity tag (hereafter cLPCAT3core) and yielded crystals with diffraction to about 3.4 Å resolution, allowing the structure determination of LPCAT3 by molecular replacement using a molecular model predicted by Tencent tFold server (https://drug.ai.tencent.com/console/cn/tfold)[38].

Overall, cLPCAT3core crystallized in the monomeric form, and residues 42–459 were resolved in each protomer (Fig. 1b and Supplementary Fig. 3a). The monomeric cLPCAT3core adapts into a bell-shaped overall structure with 11 transmembrane helices (TM1-11) and 6 shorter helices at the membrane-embed region (Ha-f) (Fig. 1b, c). No signaling peptide sequence was identified at the N-terminus of both cLPCAT3 and hLPCAT3 and both N-terminus were predicted to localize in the cytoplasmic side of the ER membrane by Phobius (https://phobius.sbc.su.se/)[39]. The transmembrane helices of cLPCAT3 gather into two wing-shape domains, an N-terminal wing (N-wing) comprising helices TM1-6, and a C-terminal wing (C-wing) comprising helices TM8-11, with the longest helix, TM7, extending from C-wing to N-wing (Fig. 1c). The helices in N- and C-wings tilt to converge at the cytoplasmic side and thus form a large cavity embedded in the exoplasmic leaflet of the ER membrane and open to the ER lumen, which is filled with loops and Ha-f (Fig. 1c, d). This wing-cavity-wing architecture resembles the folding in previously reported MOBATs, i. e. SOAT1 and DGAT1 (Supplementary Fig. 4c), where the cavity between the wings serves as a reaction chamber for the O-acyltransferase activity[34–37]. Despite their diversities in protein sequences (cLPCAT3 vs DGAT1: 13.57%; cLPCAT3 vs SOAT1: 12.86%), the proteins in structural superposition align well in their spatial arrangement, saving the exception that the membrane topology of LPCAT3 is opposite against SOAT1 and DGAT1.

The structural analysis using the program HOLLOW[40] shows a "T"-shape inner chamber enclosed by TM helices and Hd-f (Fig. 1d, e). In general, this T-shape chamber is composed of a "horizontal" tunnel parallel to the ER membrane and proximal to the cytoplasmic side, which connects with a "vertical" tunnel at the middle point (Fig. 1e). The horizontal tunnel extends from the N-wing to the C-wing and its tip in the N-wing ends at a gap between TM1 and 6 (Fig. 1d), which opens laterally to the lipid environment of the ER membrane and thus might represent a potential gate for the chamber (hereafter lateral gate). The vertical tunnel is perpendicular to the horizontal tunnel and extends to the ER lumen, with a small "side pocket" extruding in the middle (Fig. 1e). The T-shape chamber is empty in the crystal structure and the analysis on the Fo-Fc electron map failed to identify any non-proteinous density. To explore the mechanism for acyltransferase activity, further efforts in electron microscopy were conducted to determine the LPCAT3 structure in the substrate-bound states.

### The cryo-electron microscopy analysis of cLPCAT3.
Only crystals of substrate-free cLPCAT3 can diffract to a resolution beyond 4 Å and co-crystallization of cLPCAT3 with either LPCs

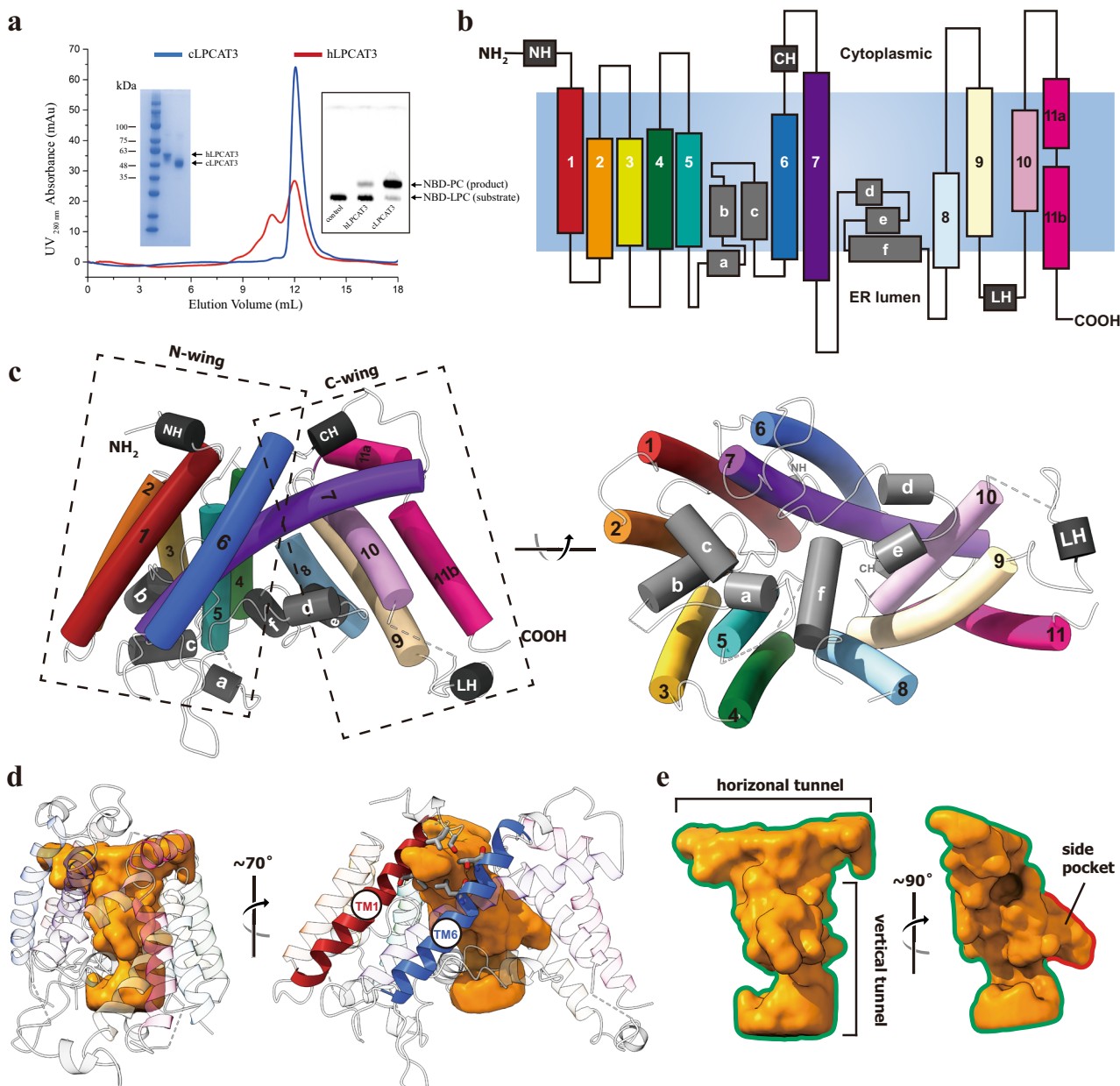

**Fig. 1 The crystal structure of monomeric cLPCAT3. a** The purification and functional assay of human and chicken LPCAT3. Blue and red curves show the size-exclusion chromatography profiles of the chicken and human LPCAT3 protein, respectively. The left inset shows the SDS-PAGE analysis on the fractions collected from the size-exclusion chromatography. The right inset shows the TLC assay on the enzymatic activities of the fractions. The experiment has been repeated three times with success. **b** Cartoon representation of cLPCAT3 monomer topology. The TM helices 1–11, N-terminal helix NH, membrane-embedded helices a-f, the cytoplasmic helix CH, and the lumenal helix LH were colored according to the same scheme as in (**c**). The cell membrane was shown as steel gray block and the orientation labeled according to the calculation using PPM server (https://opm.phar.umich.edu/ppm_server). **c** Left: the cLPCAT3 molecular model viewed parallel to the ER membrane; Right: the cLPCAT3 molecular model viewed from the lumen side of the ER membrane. **d** The T-shape chamber of cLPCAT3. The internal cavity within cLPCAT3 as determined by program HOLLOW was shown as surface in golden and the cLPCAT3 was shown as the cartoon model, with the lateral gate-forming helices TM1 and TM6 in red and blue, respectively. **e** A close view of the T-shape chamber. All structure graphs in this and the following figures were generated with PyMOL (The PyMOL Molecular Graphics System, Version 1.9 Schrödinger, LLC.) and UCSF ChimeraX (The UCSF Resource for Biocomputing, Visualization, and Informatics, version 1.2).

or acyl-CoAs failed to generate crystals capable of diffracting to 5 Å. To further explore the substrates binding and catalytic mechanism of LPCAT3, we turned to electron microscopy (EM) to determine the alternative conformations for LPCAT3. Initial efforts on single-particle analysis on cLPCAT3 samples prepared for crystallization failed to conduct a reliable 2-D classification due to the small size of monomeric LPCAT3. Extensive detergent screen in membrane solubilization and gel-filtration separation

showed that a proper proportion of cLPCAT3 protein could form an oligomer when extracted from the membrane using Lauryl Maltose Neopentyl Glycol LMNG, instead of DDM (Supplementary Fig. 2). After solubilization with LMNG, the oligomeric LPCAT3 was further separated from its monomers by size-exclusion chromatography in glyco-diosgenin (GDN) supplemented with 0.2 mM arachidonoyl CoA. Cryo-EM images of cLPCAT3 were collected using a Titan Krios transmission

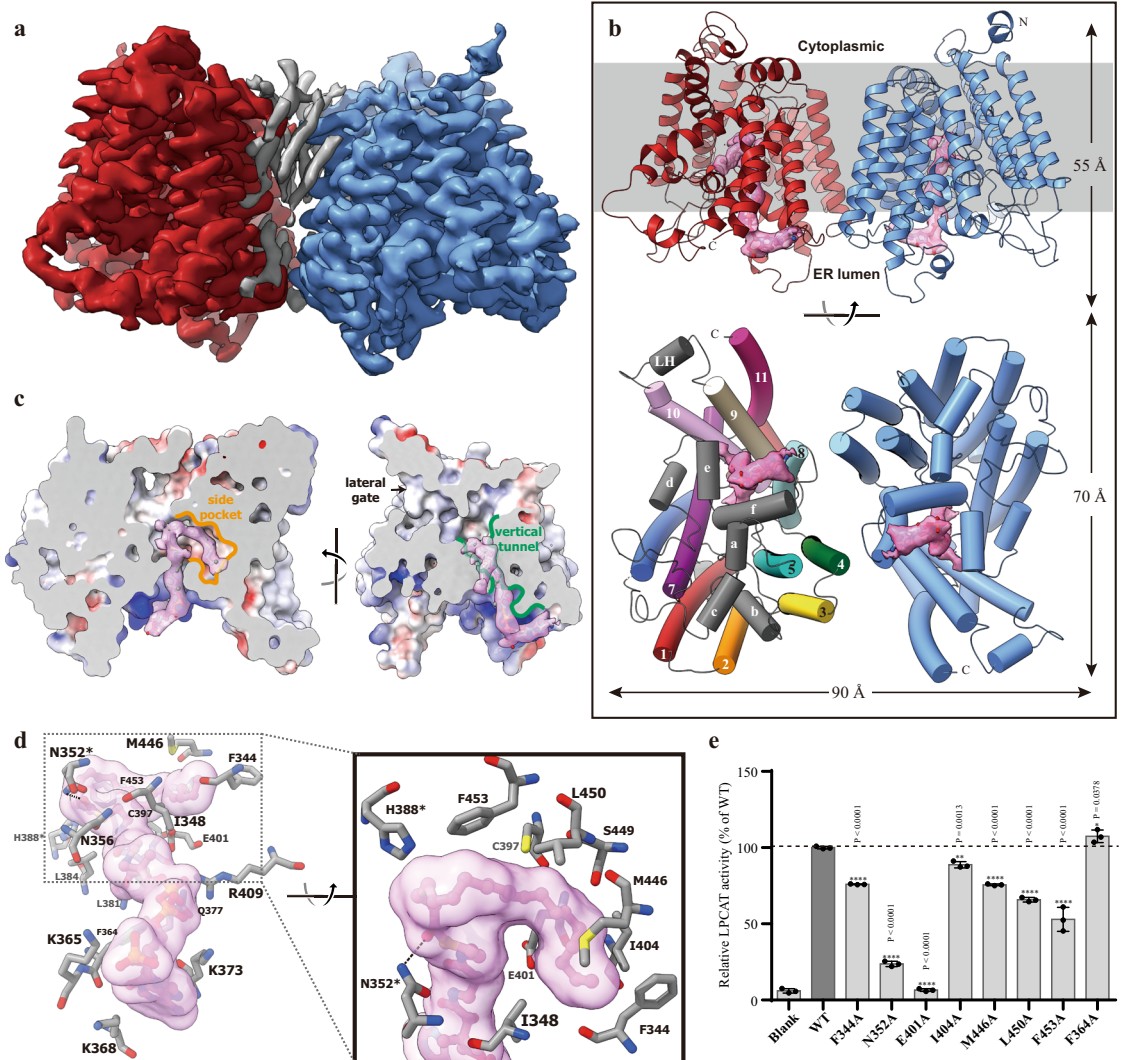

**Fig. 2 The cryo-EM structure of cLPCAT3 bound with araCoA. a** cLPCAT3 cryo-EM map viewed from the lumen side of the ER membrane. The two protomers in dimeric cLPCAT3 were shown in red and blue, respectively. The non-protein electron density at the dimeric interface was shown in gray. **b** The cartoon representation of cLPCAT3/araCoA molecular model. The non-protein electron density in the vertical tunnel and side pocket was shown in the pink surface and the araCoA molecules that fit in the density were shown as a stick-and-ball model. Upper: the cLPCAT3 viewed parallel to the ER membrane. Lower: the cLPCAT3 viewed from the lumen side of the ER membrane. One of the promotors was colored to the same scheme as in Fig. 1c for clear observation. **c** The binding of the araCoA within the vertical tunnel and side pocket. The intersecting surfaces of the T-shape chamber of cLPCAT3 were shown parallel to the ER membrane at two angles. The vertical tunnel and side pocket were highlighted in green and yellow color, respectively. **d** The interaction of araCoA with residues in the vertical tunnel and side pocket. The catalytic residue N352 was marked with an asterisk. **e** The TLC assay results on the purified wild type cLPCAT3 (dark column) and araCoA-related mutants (gray columns). The enzymatic activities of mutants were normalized as the percentage of that of wild type cLPCAT3. The results were shown as mean ± s.d.; $n = 3$ independent experiments for all mutants. One-way ANOVA with Dunnett's multiple comparisons test was applied and the 95% CI was calculated by Graphpad Prism (version 6.01), and p-values were labeled above the histogram.

electron microscope (FEI) operated at 300 kV, and data processing was performed using RELION3 and cryoSPARC[41,42]. Both 2D- and 3D-classification on the particles indicated that cLPCAT3 forms homodimer in samples (Supplementary Fig. 5). The 3D refinement using C2 symmetry generated an EM density map with an overall resolution of 3.49 Å and a resolution of about 3 Å at the core region of cLPCAT3 protomers (Fig. 2a and Supplementary Fig. 5). Similar data collections and refinement processes were conducted on the cLPCAT3 samples supplemented with 0.5 mM 1-dodecanoyl-sn-glycero-3-phosphocholine (12:0-LPC) and EM maps were generated at an overall resolution of 3.57 Å (Fig. 3a and Supplementary Fig. 6). The molecular models for the dimeric cLPCAT3/araCoA and cLPCAT3/LPC were thus built in the corresponding EM maps (Figs. 2b, 3b and

Supplementary Fig. 7), respectively, using the crystal structure of cLPCAT3core as the starting model, which showed that the cLPCAT protomers in both the araCoA- and LPC-bound state adapt into a structure similar to the monomeric crystal structure of cLPCAT3core (RMSDs among the main chains: 0.819–0.823 Å) (Supplementary Fig. 4a, b).

**The catalytic chamber and substrates bound**. A large non-protein electron density was observed in the EM map calculated with the LPCAT3 pre-incubated with 0.2 mM araCoA, which stretches along the vertical tunnel and is surrounded by helices TM8, 9, 10, and membrane-embedded helices He and Hf (Fig. 2b, c). An arachidonoyl CoA molecule was fit very well in the

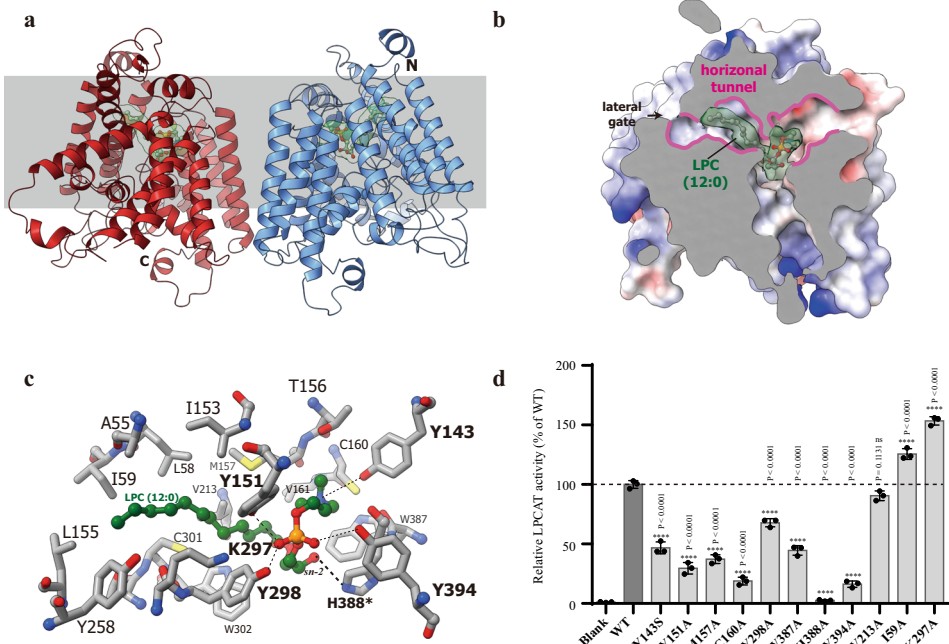

**Fig. 3 The cryo-EM structure of cLPCAT3 bound with LPC (12:0). a** The cartoon representation of the cLPCAT3/LPC molecular model as viewed parallel to the ER membrane. The non-protein electron density in the horizontal tunnel was shown in the green surface and the LPC (12:0) molecules that fit in the density were shown as a stick-and-ball model. **b** The binding of the LPC molecule within the horizontal tunnel. The intersecting surfaces of the T-shape chamber of cLPCAT3 were shown parallel to the ER membrane. The horizontal tunnel was highlighted in pink. **c** The interaction of LPC with residues in the horizontal tunnel. The catalytic residue H388 was marked with an asterisk. **d** The TLC assay results on the purified wild type cLPCAT3 (dark column) and LPP-related mutants (gray columns). The enzymatic activities of mutants were normalized as the percentage of that of the wild type cLPCAT3. The results were shown as mean ± s.d. of experiments, $n = 3$ independent experiments for all mutants. One-way ANOVA with Dunnett's multiple comparisons test was applied and the 95% CI was calculated by Graphpad Prism (version 6.01) and $p$-values were labeled above the histogram.

electron density, where the co-enzyme A moiety sits into the large blob of electron density at the ER lumenal face, while the narrow, long density corresponding to the 20:4 acyl chain stretches deep into the chamber and form a U-turn with the polyunsaturated scaffold of arachidonoyl group to enter the side pocket (Fig. 2c), allowing the vertical tunnel accommodates the oversized arachidonoyl group and saving the space in the horizontal tunnel for the acyl acceptor LPC. The cytoplasmic end of the vertical tunnel is lined with positively charged amino acids and thereby provides a docking surface for the coenzyme A moiety, where the binding of CoA is stabilized by the electrostatic interactions among the pyrophosphate section of CoA and basic residues R409, as well as K365, 368, and 373 (Fig. 2d). The araCoA extends into the T-shape chamber and turns at the joint point near H388 to a "side pocket" with its double-bonds in the acyl chain (Fig. 2c), and the carbonyl oxygen of the arachidonoyl pointing to the conserved residue N352, implying its role as protonation reagent in the hydrolysis of thioester bond. When N352 was mutated, the cLPCAT3 show an impaired acyltransferase activity, confirming its critical role in the reaction (Fig. 2e). To hold the acyl chain, the side pocket is in general lined with hydrophobic residues, e.g., F344, I348, I404, M446, and L450, and the functional assay showed that the mutations on those sites could reduce the enzymatic activities to various degrees (Fig. 2e). Unexpectedly, the E401A mutation fully abolished the acyltransferase activity (Fig. 2e), implying the importance of E401 in maintaining the binding of the acyl-CoA molecule.

In the cryo-EM map for cLPCAT3/LPC was identified a large non-protein electron density located in the horizontal tunnel with no overlap with the araCoA electron density found in the EM map for cLPCAT3/araCoA (Fig. 3a, b). The C12:0-LPC molecule fits well in the density with the phosphorylcholine head

positioned in the C-wing side and stabilized by the interactions with a series of hydroxy groups from tyrosine Y143, Y151, Y298, and Y394 (Fig. 3c). When the interactions were impaired by the mutations of those tyrosine residues, the enzymatic activities of cLPCAT3 reduced significantly (Fig. 3d). The C12:0-acyl at sn-1 of the LPC extends to the N-wing side of the horizontal tunnel and ends near to the lateral gate formed by TM1 and 6 (Figs. 1d and 3b), where the binding of acyl chain is mediated by its hydrophobic interaction with residues L58, I59, Y151, I153, L155, and V213. The sn-2 hydroxy of the glycerol backbone sits in close proximity to the catalytic residue H388 (Fig. 3c), ready to be activated into a nucleophilic reagent through the deprotonation by the imidazolyl of the histidine.

**The structural implication of the acyltransferring mechanism.** Previous structural studies on lipid acyltransferases, e.g. DGAT1 and SOAT1, revealed the entrance and binding of acyl donor molecules in enzyme, and the catalytic mechanism were proposed combined the experimental molecular model and putative binding pocket for acyl receptor[34–37]. In our studies on LPCAT3, the acyl donor and receptor molecules bound are visible in two EM maps with an adjacent but separate spatial arrangement, allowing a straightforward analysis of the acyl transferring process. Since the protein backbone in the structures of LPCAT3 bound with LPC and araCoA are in general consistent, the two EM models for cLPCAT3 were merged to generate a working model for LPCAT3 (Fig. 4a, b). The T-shape chamber with the side pocket could allow the enzyme to capture the acyl donor and receptor simultaneously, where the carbonyl oxygen of the acyl donor and the sn-2 hydroxy of the acyl receptor could be activated by N352 and H388, respectively (Fig. 4c). The reaction could be then continued by the formation of the ester bond between the oxygen

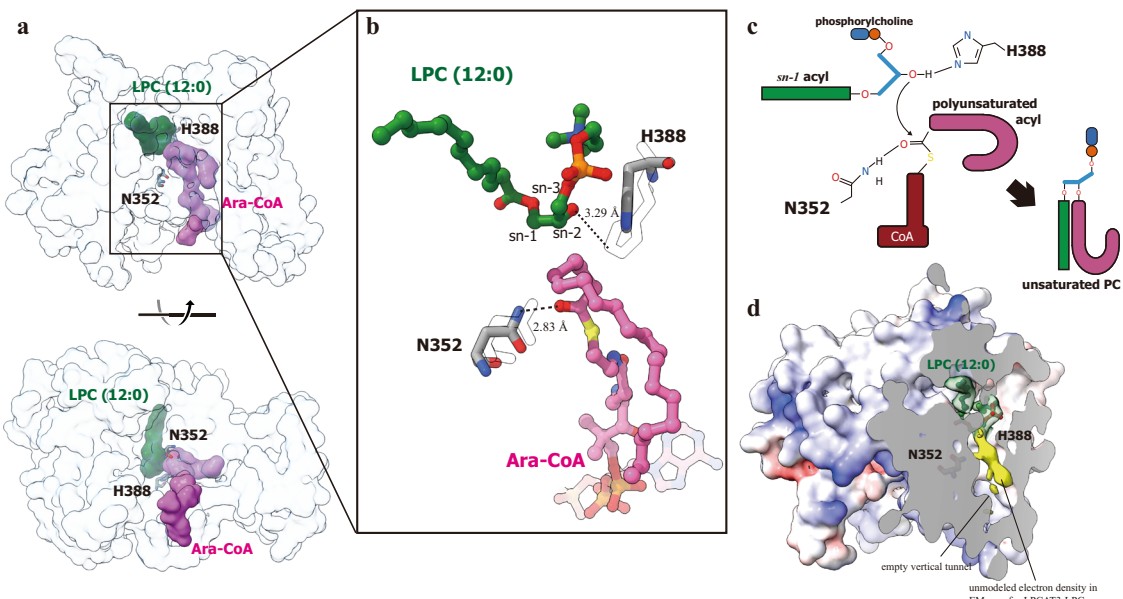

**Fig. 4 The merged molecular model for cLPCAT3 in dual-substrates bound state. a** The model of cLPCAT3/LPC protomer was superposed with the model of cLPCAT3/araCoA protomer which is shown in transparent blue surface for cLPCAT3 and pink surface for araCoA, and the LPC molecule was then shown as green surface at its corresponding position in cLPCAT3/araCoA to generate the merged molecular model. **b** A close view on the spatial organization of the two substrates of acyl transferring reaction, as well as the catalytic residues, H388 and N352. The residues H388 and N352 were shown in stick model with the corresponding residues in cLPCAT3/LPC model shown as transparent sticks. The araCoA and LPC molecules were shown as stick-and-ball models. **c** The putative mechanism for the acyl transferring by LPCAT3. **d** The unmodeled electron density revealed in the EM map of cLPCAT3/LPC. The surface model of cLPCAT3/LPC was dissected to show the LPC electron density (in green) and unmodeled density (in yellow).

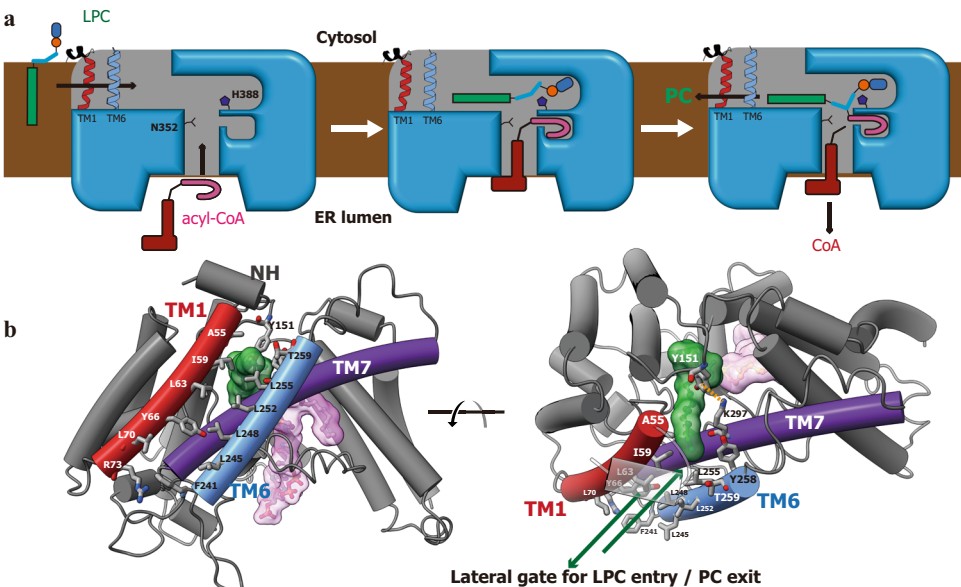

**Fig. 5 The working model for LPCAT3. a** The cartoon representation about the enzymatic operation of LPCAT3. The whole reaction cycle includes three states: substrate-capture (left), acyl transferring (middle), and product release (right). **b** Close views on the lateral gate for LPC entry and PC exit from two angles. The LPC molecule bound was shown as a stick-and-ball model covered by green surface. The residues from TM1 and 6 lining the lateral gate were shown and labeled. The Y151-K297 pair functions as a lock for the gate through their cation-aromatic interaction (orange dot line).

of sn-2 hydroxy and carbon of protonated carbonyl, followed by the breakage of the thioester bond to release the PC product from the intermediate. In the end, the CoA molecule thereby formed exits from the vertical tunnel into the ER lumen, while the PC molecule might enter the ER membrane through the lateral gate (Fig. 5a, b). The enzymatic activity of LPCAT3 was improved by the mutations on the residues at the lateral gate, e. g., I59A (Figs. 3d and 5b), implying the gating function of I59 for limiting

the passage of PC/LPC. An even higher rise in enzymatic activity was found on cLPCAT3 K297A, which could relieve the steric hindrance from the cation-aromatic interaction between Y151 and K297 (Fig. 5b).

When studying DGAT1 in the acyl donor-bound state, Sui et al. showed a lipid-like density co-localized in the reaction chamber which might represent the DAG molecule as the acyl receptor[36]. In our EM map for acyl donor-bound state

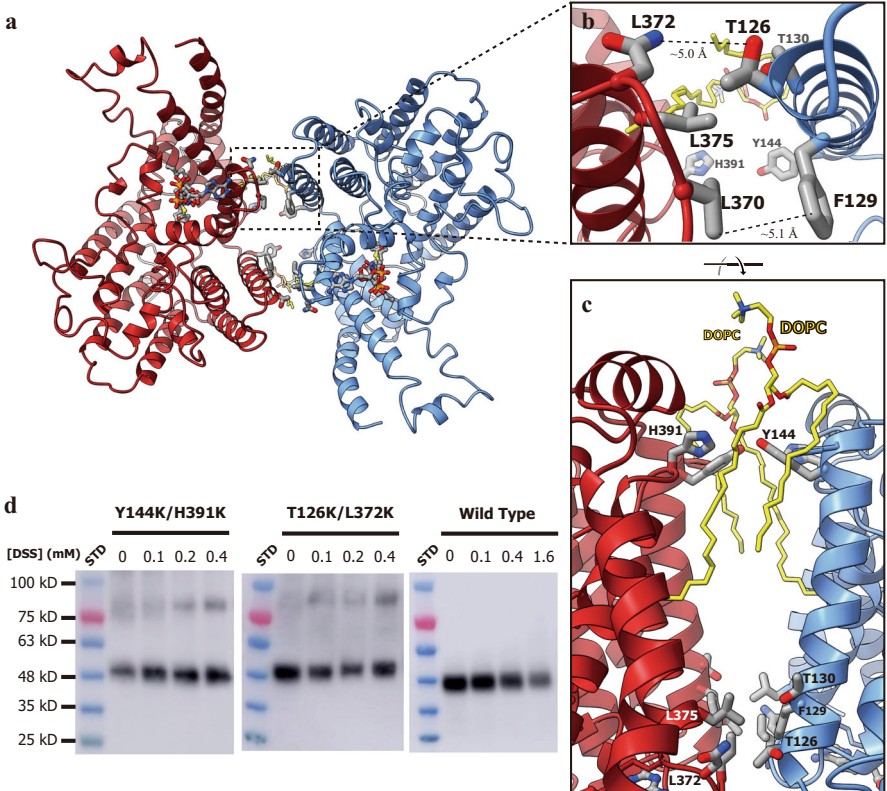

**Fig. 6 The non-contact dimer of LPCAT3. a** The molecular model of LPCAT3/araCoA viewed from the lumen side of the ER membrane. The residues and DOPC molecules at the non-contact dimeric interface, as well as the araCoA molecules, were shown as stick model. **b**, **c** A close view of the dimeric interface at two angles. **d** The crosslinking assay on membrane-bound cLPCAT3. The wild type cLPCAT3 and the double mutants of the dimeric interface, T126K/L372K and Y144K and H391K, were treated with the crosslinking reagents DSS at the concentrations as indicated before solubilized from the membranes of the Expi293 cells overexpressing the corresponding proteins. Shown were the representative results from three independent experiments.

(cLPCAT3/araCoA), the binding cavity for LPC holds some small blobs of electron density and no continuous density was visible there (Supplementary Fig. 9b). Interestingly, despite the empty vertical tunnel in our EM map for acyl receptor-bound state (cLPCAT3/LPC), a long, lipid-like density was revealed in the side pocket which meets the electron density of LPC at the joint point near to H388 (Fig. 4d). Manually modeling the acyl of LPC into the density failed to generate well-fit models, suggesting this lipid-like density might not represent an alternative conformation of LPC molecules bound (Supplementary Fig. 9a). Based on the absence of density corresponding to the CoA moiety, we speculated that this unmodeled density, in combination with the large density for LPC, might represent the PC product partially bound and co-purified with the protein. Further studies should be conducted to probe the identity of this density and the exit mechanism for remodeled PC.

**The non-contact dimer of LPCAT3.** Although the electron microscopy analysis showed that cLPCAT3 could be purified as a dimer with relatively mild detergent LMNG, cLPCAT3 shows an unusual dimerization conformation without any physical contact between two protomers. Only seven residues stay at a distance less than 5 Å from the neighboring protomer and the shortest distance between two protomers is about 4.4 Å (Fig. 6b), insufficient for stable dimer maintenance. The mutations of residues at the close point of the dimer interface, such as T126K/L372K double mutation, cannot disrupt the dimer, implying the limited involvement of the residue sidechains in the dimerization of LPCAT3 (Fig. 6d). The cryo-EM map analysis on cLPCAT3/araCoA revealed the non-protein electron density sandwiched by

cLPCAT3 protomers (Fig. 2a) and two 1,2-dioleoyl-sn-glycero-3-phosphocholine (DOPC) molecules were well modeled into the large blobs of electron density (Fig. 6c and Supplementary Fig. 7). This non-contact dimerization might be mediated by those intermediary lipid molecules and the relatively harsh detergents used in the crystallization trial deprived the cLPCAT3 of the lipids bound, resulting in the monomeric biological unit in cLPCAT3 crystals. To probe the natural oligomeric state of LPCAT3 in the membrane, cross-linking assays were conducted on membrane preparations from the Expi293 cells overexpressing cLPCAT3. The wild type cLPCAT3 showed weak dimer bands in SDS-PAGE upon the treatment of the amine-crosslinking reagents used, probably resulting from the lacking of primary amine group pairs at the dimeric interface. Double mutant cLPCAT3$^{T126K/L372K}$ and cLPCAT3$^{Y144K/H391K}$ introduced primary amine groups at the interface and can be well crosslinked to form a dimer in membrane-bound state, indicating the existence of natural LPCAT3 dimer in the membrane (Fig. 6d). This unstable interface might keep LPCAT3 in a dynamic balance between the monomeric and dimeric states and the biological relevancy for this oligomerization remains to be elucidated.

## Discussion

LPCAT3 plays a major role in modulating the level of poly-unsaturated PLs in the cellular membrane system, especially that of arachidonoyl-containing PCs[16,43,44]. The substrate preferences of LPCAT3 on the polyunsaturated acyl chain of the acyl donor and the phosphocholine group of the acyl receptor, however, are yet to be elucidated. In the T-shape catalytic chamber revealed in our LPCAT3 structure, the length of the vertical tunnel is limited

and could only accommodate the CoA moiety with a short acyl group. The side pocket substructure extruding from the vertical tunnel expands the space for acyl-CoA, but its near-to-90° stretching routine from vertical tunnel might be unfavorable for capturing the chemical structure of saturated acyl chain, such as oleoyl. Although the C–C single bonds render the saturated carbon chain with more flexibility, the stable staggered rotamers of C–C single bond tend to keep the saturated acyl in an extended, all-trans configuration, making the large-angle bending of the carbon chain energetically unfavorable[45]. The all-cis configuration in polyunsaturated acyl chain, such as a 20:4 acyl, could help carbon chain adapt into a ring-shape structure fitting the side pocket very well, while a saturated acyl chain with moderate length, such as 18:0 or 16:0, would suffer from stereo-hindrance for their straighter conformation and thus be catalyzed with significantly poorer efficiency. This substrate preference mechanism is supported by the substrate preference studies by Zhao et al, where the LPCAT3 showed the highest acyltransferase activities using long polyunsaturated acyl-CoA (20:4 and 18:2) as acyl donor, while a preference on shorter carbon chain was observed on the saturated acyl-CoA (12:0 > 16:0 > 18:0)[12].

In precedent biochemical assays on LPCAT3, it was shown that it catalyzes acyl transferring reaction with a hill coefficient of about 2 (from 1.4 to 3.7 for different acyl-CoAs), implying the existence of oligomeric LPCAT3[43]. Although crystallized as a monomer, the cLPCAT3 forms dimer in both membrane-bound state and subsequent EM analysis in detergents. Different from dimeric DGAT1 or SOAT1[35,37], LPCAT3 dimerizes at helices TM3, 4, and 8, which align well with the distal region from the dimeric interface in DGAT1 and SOAT1 dimer (helices TM2, 3, and 7 of DGAT and SOAT1, respectively, see Supplementary Fig. 8b). This alternative architecture of LPCAT3 dimer exposes its lateral gate to the ER membrane environment and thus might allow a fast substrate/product exchange during the reaction.

According to the sequence analysis and the structures solved, the topology of cLPCAT3 shows an ER lumen-facing gate for the acyl-CoA substrate, an opposing direction against recently determined structures of MBAOT members, e.g., SOAT1, DGAT1, and HAAT[34–37,46,47]. The possibility cannot be ruled out that LPCAT3 adopts a membrane topology opposite to that we proposed, since an exoplasmic/ER lumen localization of N-terminal without signaling peptide could be found in some cases, e.g., the bacterial MOBAT DltB[48]. Therefore, we conducted an on-cell protease digestion assay using the sequence-specific proteases HRV-3C to treat the cLPCAT3-overexpressing cells to probe the accessibility of the N- and C-terminus of LPCAT3 (Supplementary Fig. 10a). The results showed that, when the proteases were applied to intact cells, the protease recognition sequence at the C-terminal of cLPCAT3 can be cleaved but that at the N-terminal of cLPCAT3 was only slightly cleaved (Supplementary Fig. 10c), indicating a cytoplasm-localized N-terminal and an exoplasmic/ER lumen localization of C-terminal for cLPCAT3 as suggested in our structural model. To test if the proteases used in this assay have sufficient proteolytic activities on the cLPCAT3 in the membrane-bound state, a control experiment was conducted on the membrane debris prepared by sonication-disruption of the cells overexpressing cLPCAT3, and the results confirm that the protease recognition sequence at both N- and C-terminus of cLPCAT3 can be cleaved by the proteases after cell disruption (Supplementary Fig. 10c). We believe further analysis should be conducted to reliably establish the topology of LPCAT3 since the assay on the overexpressed proteins might introduce unexpected artifact due to the unnatural expression level and localization. However, a further question is raised about the substrate availability if our current hypothesis is true, since

the arachidonoyl CoA are mainly synthesized in the cytoplasm by long-chain acyl-CoA synthetase such as ACSL4[49,50]. The transportation of arachidonoyl CoA across the ER membrane is thus required for the lipid remodeling catalyzed by LPCAT3, and the mechanism underlying this process is yet to be elucidated.

In a conclusion, we determined the structures for the critical enzyme in the phospholipid remodeling, LPCAT3, in its apo-, LPC-bound, and arachidonoyl CoA-bound states. In those structures, the binding pockets for both acyl-donor and receptor were mapped and the arrangement of the catalytic residues provide insights into the mechanism of acyl transferring reaction. The T-shape chamber revealed in our structures, in combination with the lateral gate and side pocket, show the structural basis for the substrate preferences of LPCAT3 and substrate/product exchange process.

## Methods

**Protein expression and purification.** The coding cassettes for *Gallus gallus* LPCAT3 (cLPCAT3, Uniprot ID A0A1L1RNG8, https://www.uniprot.org/uniprot/A0A1L1RNG8) and *Homo sapiens* (hLPCAT3, Uniprot ID Q6P1A2, https://www.uniprot.org/uniprot/Q6P1A2) were synthesized and cloned into pFastBac Dual vector and pcDNA3.4 vector modified to introduce a Human Rhinovirus 3C Protease cleavage site followed by a Twin-Strep tag at the C-terminal of LPCAT3s.

Bac-to-Bac Baculovirus Expression System (Invitrogen) was used to express the LPCAT3s protein for crystallization trial. The recombinant baculovirus encoding cLPCAT3 or hLPCAT3 was generated and infected into *Spodoptera frugiperda* cell line Sf9 for overexpression. The infected cells were further cultured for 60–70 h and collected by centrifugation (1500 × *g*, 10 min, 4 °C). All the purification procedures were carried out at 4 °C. Firstly, the cell pellets were homogenized in a Low Salt buffer (10 mM NaCl, 10 mM HEPES, pH 7.5) containing 1 mM Tris(2-carboxyethyl)phosphine (TCEP) and 1 mM PMSF (phenylmethanesulfonyl fluoride). The cell debris was harvested by centrifugation (45,000 × *g*, 25 min, 4 °C) and homogenized in a High Salt buffer (1 M NaCl, 25 mM HEPES, pH 7.5) containing 1 mM TCEP, 1 mM PMSF, 5 mM MgCl₂, 0.1 mg mL⁻¹ DNase I. The membrane fraction was collected again by centrifugation. The pellet was resuspended and homogenized in Lysis Buffer (150 mM NaCl, 20 mM HEPES, pH 7.5, 10% (v/v) Glycerol) in the presence of 1 mM TCEP, 5 mM MgCl₂, 0.1 mg mL⁻¹ DNase I, 1× protease inhibitor cocktail (MCE). The membrane was solubilized with 1% (w/v) n-Dodecyl-β-D-Maltopyranoside (Exceedbio) at 4 °C for 2 h. The solubilization slurry was clarified by centrifugation at 45,000 × *g* for 45 min, and the LPCAT3s proteins were further purified by affinity chromatography using Strep-Tactin resins (IBA). The resins was washed by Lysis Buffer containing 1 mM TCEP, 2.4 mM n-undecyl-β-D-maltopyranoside (Anatrace) and the proteins were eluted with same buffer supplemented with 5 mM D-desthiobiotin. The eluate was concentrated to about 5 mg mL⁻¹ and then trypsin was added to a concentration of 10 μg mL⁻¹. After incubating for 30 min at 4 °C, the trypsin-treated sample was subjected to size-exclusion chromatography using Superdex 200 Increase 10/300 GL column (GE Healthcare) in a mobile phase containing 150 mM NaCl, 20 mM HEPES, pH 7.5, 1 mM TCEP, and 1.2 mM UM. The fractions of elution peak were pooled and concentrated to about 10 mg mL⁻¹ for crystallization.

Expi293 cells were used to express proteins for cryo-EM sample preparation. When cells density reached ~2.0 × 10⁶ cells mL⁻¹, plasmids with LPCAT3 constructs were transiently transfected into cells using PEI MAX (Polysciences), and the transfected cells were harvested after 72 h. The cells were collected by centrifugation at 3000 × *g* for 10 min and resuspended in Lysis Buffer supplemented with 5 mM MgCl₂, 0.1 mg DNase I, 1 mM PMSF. The cell membrane was solubilized with Lysis Buffer supplemented with 1% (w/v) lauryl maltose neopentyl glycol (LMNG, Anatrace), 5 mM MgCl₂, 0.1 mg mL⁻¹ DNase I, 1 mM TCEP, 1× protease inhibitor cocktail at 4 °C for 2 h. After centrifugation at 45,000 × *g* for 45 min at 4 °C, the supernatant was subjected to the affinity chromatography using Strep-Tactin resins and the LPCAT3 proteins elated by the Lysis Buffer with 1 mM TCEP, 0.01 % GDN (w/v, Anatrace), and 5 mM D-desthiobiotin, pH 7.5. The Twin-Strep tags of LPCAT3 proteins were removed with 3C protease, and the LPCAT3 protein was further purified by size-exclusion chromatography using Superdex 200 Increase 10/300 GL column in a mobile phase containing 150 mM NaCl, 20 mM HEPES, pH 7.5, 1 mM TCEP and 0.01% (w/v) GDN. To purify the cLPCAT3 in complex with arachidonoyl-CoA and lysophosphatidylcholine (12:0), 0.1 mg mL⁻¹ arachidonoyl-CoA lithium salt (Arac-CoA-Li, Sigma) or 0.1 mg mL⁻¹ lysophosphatidylcholine (12:0, Avanti Polar Lipids, Inc.) was added during the membrane solubilization and the affinity chromatography, respectively. The elution fractions corresponding to the dimeric LPCAT3 were pooled and re-subjected to size-exclusion chromatography using Superdex 200 Increase 10/300 column with the same mobile phase. The fractions of elution peak were then pooled and concentrated to approximately 7 mg mL⁻¹, with 0.2 mM Arac-CoA-Li or 0.5 mM Lyso-PC (12:0) supplemented, respectively.

**Crystallization and X-ray data collection**. Crystallization trails was extensively performed on both hLPCAT3 and cLPCAT3 using both vapor diffusion and liquid cubic phase configurations. The crystals of trypsin-digested cLPCAT3 were grown by vapor diffusion at 16 °C and harvested in 14–28 days by directly flash freezing into liquid nitrogen. The crystals with the best diffraction capacity were grown by mixing trypsin-digested cLPCAT3 with a solution containing 50 mM Magnesium acetate (Sigma), 100 mM MES, pH 6.0, 25% (v/v) PEG 400, and 1% (v/v) Formamide. Crystals were harvested and cryoprotected by soaking for 30 s in a solution containing 32% PEG 400, 100 mM MES, pH 6.0, 50 mM magnesium acetate, and then flash-frozen into liquid nitrogen. The diffraction data were collected at the Shanghai Synchrotron Radiation Facility (beamline stations BL18U1 and BL19U1) with a wavelength of 0.978530 Å and a temperature of 100 K.

**Electron Microscopy sample preparation and data collection**. The cryo-grids were prepared using Thermo Fisher Vitrobot Mark IV operated at 8 °C with 100% humidity. 3.2 μL of concentrated protein solution with arachidonoyl-CoA (ara-CoA) or lysophosphatidylcholine (LPC) were applied to glow-discharged holey carbon grids (Quantifoil R1.2/1.3, Au, 300 mesh). The protein sample was incubated for 10 s and then blotted with filter paper (Waterman) for 1.5 s. The grids were then plunged into the liquid ethane cooled with liquid nitrogen. The cryo-grids were firstly checked by 200 kV Talos Arctica (FEI) equipped with a Falcon III detector (Thermo Fisher Scientific) and a dataset was collected to confirm the sample quality and generate an initial model. For super resolution data collection, micrograph stacks were automatically collected with EPU software on Titan Krios (FEI) operated at 300 kV equipped with a K3 Summit direct electron detector (Gatan), a Quantum energy filter (Gatan) and Cs corrector (Thermo Fisher), functioning in zero-energy-loss mode. The Data was collected at a nominal magnification of ×105,000 (corresponding to a physical pixel size of 0.86 Å), with a defocus range between −1.0 and −2.5 μm. The dose rate was set to ~18.2 electrons/Å2/s and the total exposure time was 2.7 s, resulting in a total dose of 50 electrons/Å2, fractionated into 32 frames.

**Electron microscopy data processing**. All the data processing for cryo EM was perform using program RELION-3[51] and cryoSPARC[41] (version 3.2.0). Two datasets of 11,169 and 13,687 micrographs were collected for the cLPCAT3 supplemented with araCoA and LPC (12:0), respectively. The motion correction was performed using MotionCorr2 (version 1.1.0) with 6× 5 patches[52], and the CTF parameters was estimated using Gctf (version 1.1.8) to discard the micrographs with poor statistics[53]. The particles were automatically picked using Gautomatch-v0.56 (developed by Kai Zhang) with a 2D reference from 200 kV EM data.

For LPCAT3 with araCoA, 4,662,400 particles were firstly extracted with 8× binning and the 2D classifications excluded particles with poor quality. 3,700,472 particles were then re-extracted with 2× binning and subjected to the global 3D classification with a reference from 200 kV EM data processing. 1,640,907 particles were selected and the best EM map was low-passed to 10 Å, 30 Å, and 60 Å, then all four maps were combined to generated a multi reference for further 3D classification[54]. After two rounds of 3D classification, 837,336 particles corresponding to suitable classes were re-extracted without binning and subjected to processing with cryoSPARC. 292,613 particles were subjected to Ab-initio Reconstruction and four models were generated. All particles were pooled into Heterogeneous Refinement, and after six cycles of classification, 282,382 particles belonged to the best class were subjected into Non-uniform Refinement (Legacy) with with C2 symmetry and an adaptive solvent mask, thereby yielding a map with an overall resolution of 3.49 Å. Local resolution estimation was performed by cryoSPARC and all the resolutions were estimated using the gold-standard Fourier shell correlation 0.143 criteria with the high-resolution noise substitution[55,56].

For LPCAT3 with LPC, 8,073,719 particles were firstly extracted with 8× binning and the 2D classifications excluded particles with poor quality. 5,925,352 particles were re-extracted with 2× binning and subjected to 3D classification. After four cycles multi-reference 3D classification, 652,955 particles corresponding to suitable classes were re-extracted without binning and further processed with cryoSPARC. Ab-initio Reconstruction were used to generated four models and the same particles were pooled into Heterogeneous Refinement. After four rounds of iterations, 223,896 particles were selected and Non-uniform Refinement was conducted with C2 symmetry and an adaptive solvent mask, thereby yielding a map with an overall resolution of 3.57 Å. Local resolution estimation was performed by cryoSPARC and all the resolutions were estimated using the gold-standard Fourier shell correlation 0.143 criteria with the high-resolution noise substitution.

**Model building and refinement**. For the X-ray diffraction data processing on cLPCAT3core crystals, an initial model was built using the Protein Structure Prediction by tFold (https://drug.ai.tencent.com/console/en/tfold). The crystallography dataset was processed using program HKL-2000 (HKL-2000 version 716.1), then solved by molecular replacement in Phaser (packaged in Phenix version 1.19.2-4158) and was refined with program Coot (Coot version 0.9.5,)[57]. and Buster (BUSTER 2.10.3, Cambridge, United Kingdom: Global Phasing Ltd.).

The model building for cryo EM data was performed with program Coot using the crystal structure of cLPCAT3core as the starting model. The initial model was iteratively adjusted and refined with the Real-space refinement in Phenix package[58] and Coot The araCoA or LPC ligands were found and refined by the program LigandFit and Real-space refinement in Phenix package, respectively. The percentage of Ramachandran outliers is 0.7%, and the percentage value of Ramachandran allowed and favored are 7.7% and 91.5%, respectively.

**TLC-based activity assay**. The lysophosphatidylcholine acyltransferase (LPCAT) activity of LPCAT3 was determined by monitoring the appearance and the intensity of the fluorescent band generated from the incorporation of the arachidonoyl group from araCoA into NBD labeled Lyso-PC (12:0, Avanti Polar Lipids)[12]. A general reaction system contained 2 mM DDM, 150 mM NaCl, 20 mM HEPES, pH 7.5, 1 mM TCEP, 8 μM NBD-Lyso PC, 19 μM Ara-CoA-Li in a total volume of 200 μL. The reaction was initiated by adding 1.2 μg purified proteins and incubated for 15 min at room temperature, the reaction was terminated by adding 600 μL of chloroform/methanol (1:1) mixture. After centrifugation at 2500 × g for 10 min, the organic phase was dried by high pure nitrogen and redissolved in 30 μL chloroform or 6 μL DMSO. The mixture was separated by thin layer chromatography (TLC) with chloroform/methanol/H2O (65:25:4) and imaged by Amersham Typhoon (GE Healthcare).

**Crosslinking assay on membrane-bound cLPCAT3**. Expi293 cells were transfected with expression vector carrying cDNA for wild-type cLPCAT3 or its mutants and harvested after 60 h. Cells were disrupted by sonication and membrane debris was collected by centrifugation at 45,000 × g for 15 min at 4 °C. The pellets were resuspended and homogenized in Lysis Buffer (150 mM NaCl, 20 mM HEPES, pH 7.5, 10% (v/v) Glycerol) in the presence of 1 mM TCEP, 5 mM MgCl2, 0.1 mg mL−1 DNase I, 1× protease inhibitor cocktail (MCE). DSS (disuccinimidyl suberate, ThermoFisher Scientific Inc.) was used as crosslinking regent. The total volume of each reaction was 50 μL and the same volume of DSS in DMSO at different concentrations was added and the pure DMSO was also added as a negative control. To initiate the crosslinking reaction, DSS dissolved in dimethyl sulfoxide (DMSO) was added into the membrane preparation to various concentrations as indicated in Fig. 6d. The reaction mixture was incubated at room temperature for 30 minutes and then quenched by adding Tris-HCl (pH 7.5) to a final concentration of 50 mM. The mixture was solubilized with 1% (w/v) Digitonin (Merck KGaA) at 4 °C for 2 h, The solubilization slurry was clarified by centrifugation and the supernatants were subjected to SDS-PAGE followed by Western blot analysis. Mouse anti-Strep II-Tag multi antibody (ABcolonal Inc.) was used as the primary antibody and HRP coupled anti-mouse antibody as the secondary antibody. The blotting images were photographed with Amersham Imager 600 (GE Healthcare).

**On-cell protease digestion assay**. The coding cassettes of cLPCAT3 was cloned into pcDNA3.4 vector with a N-terminal Flag tag and a C-terminal Twin-Strep tag, both spaced by a GS linker (SGGSGGGSGGG) and a HRV-3C protease recognition site, respectively (Flag-cLPCAT3-Strep). Plasmids were transiently transfected into Expi293 cells using PEI MAX (Polysciences) and the cells were cultured for a further 48 h. Equal aliquots of cell culture were used in on-cell protease digestion assay and control group, respectively. For on-cell protease digestion assay, cells were collected by centrifugation at 200 × g for 3 min. The supernatant was removed and the cells were resuspended in fresh medium (Cell wise) with gentle agitation, and 0.08 mg mL−1 HRV-3C was added into the suspension of the cells overexpressing Flag-cLPCAT3-Strep. The cell-protease mixture was incubated at room temperature for 20 min and the digestion was stopped by washing the cells three times in PBS supplemented with 0.1 mg mL−1 Iodoacetamide. The cells were then resuspended in Lysis Buffer in the presence of 5 mM MgCl2, 0.1 mg mL−1 DNase I, 1× protease inhibitor cocktail, 0.1 mg mL−1 iodoacetamide and DDM was used to solubilize the membrane protein. For the control group, the cells were collected and resuspended in Lysis Buffer in the presence of 5 mM MgCl2, 0.1 mg mL−1 DNase I, 1× protease inhibitor cocktail. Cells were then disrupted by sonication and then 0.08 mg mL−1 HRV-3C protease was added into the lysis from the cells overexpressing Flag-cLPCAT3-Strep. The lysis-protease mixture was incubated at 4 °C for 20 min and the membrane debris were pelleted by centrifugation. The membrane debris were further washed by resuspending in Lysis Buffer supplemented with 0.1 mg mL−1 iodoacetamide and collected by centrifugation to remove residual proteases. The debris was resuspended in Lysis Buffer in the presence of 5 mM MgCl2, 0.1 mg mL−1 DNase I, 1× protease inhibitor cocktail, 0.1 mg mL−1 Iodoacetamide. DDM was added to solubilize the membrane protein. The solubilization slurries from both the experimental and control group were clarified by centrifugation and the supernatant was subjected to SDS-PAGE followed by western blot. We used antibodies against the fusion tags (Mouse anti Strep II-Tag mAb (ABclonal, AE066) and DYKDDDDK Tag Mouse mAb (Cell Signaling, #8146)) to detect the cLPCAT3 overexpressed and the proteolysis efficiency was estimated by the reduction of the blotting bands intensity, we also use GAPDH Monoclonal Antibody Mouse Monoclonal (proteintech, Catalog number: 60004-1-Ig |CloneNo.: 1E6D9) to evaluate the cell quantity. The secondary antibody is Anti-mouse IgG, HRP-linked Antibody (Cell Signaling, #7076). All the antibodies were diluted

(1:10,000) by skim milk in TBST in prior to use. Images were captured by Amersham Imager 600.

**Reporting summary**. Further information on research design is available in the Nature Research Reporting Summary linked to this article.

## Data availability

The crystallographic data in this study have been deposited in Protein Data Bank (PDB) under accession code 7EWT. The cryo-EM models have been deposited in the EMDB and PDB with the accession codes 7F3X and 7F40. The cryo-EM maps have been deposited in the Electron Microscopy Data Bank (EMDB) with accession codes EMD-31442 and EMD-31443. All the other data are available from the corresponding authors upon reasonable request. Source data are provided with this paper.

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

## Acknowledgements

The authors thank Drs. Ming Zhou, Tingbo Ding, Sheng Wang, and Lijun Wang (for scientific discussion); Yijun Gu, Rijing Liao, and Mi Cao (for assistance with the data analysis). This work is supported by National Key Research and Development Program of China (2018YFC1004704 and 2017YFC1001303, Y. Cao), National Natural Science Foundation of China (82072468, Y. Cao), the Shanghai Science and Technology Committee (20S11902000, Y. Cao), SHIPM-pi fund (No. JY201804, Y. Cao) from Shanghai Institute of Precision Medicine, Ninth People's Hospital, Shanghai Jiao Tong University School of Medicine. This work is also supported by Innovative Research Team of High-level Local Universities in Shanghai (SSMU-ZLCX20180600, M. Lei and Y. Cao) and the Special Project of Shanghai Synchrotron Radiation Facility (SSRF) BL18U1 (2020-NFPS-ZD-000146, L. Zhou).

## Author contributions

Y. Cao, L. Zhou, and X.C. Jiang initiated the study. Y. Cao, Q. Zhang, M. Lei, J. Zhao, and A. Qin designed research. Y. Cao, Q. Zhang, and X.C. Jiang wrote the paper. Q. Zhang, Y. Chen, L. Jan, and K. Hu. performed the purification and EM sample preparation. Q. Zhang, B. Rao, S. Li, Y. Shen, and Y. Xia collected and analyzed the data. Q. Zhang conducted the biochemical assay. Y. Cao and D. Yao determined the structure. We thank the staff members of the Electron Microscopy System and Mass Spectrometry System at Shanghai Institute of Precision Medicine for providing technical support and assistance in data collection. SSRF beamlines BL18U1 and BL19U1 are used for X-ray crystallography data collection.

## Competing interests

The authors declare no competing interests.
