## [Peer Review File · Nature Communications]

The structural basis for the phospholipid remodeling by lysophosphatidylcholine acyltransferase 3REVIEWER COMMENTS

Reviewer #1 (Remarks to the Author):

Qing Zhang et al solved the crystal structure of apo-LPCAT3 and cryo-EM structures of acyl-donor and acyl-acceptor bounded states of LPCAT3. It is the first time for an acyl-acceptor substrate to be experimentally visualized in 3D structure for MBOATs that take lipids as acyl group acceptors in acyl transfer reaction. The results in this manuscript will provide important insights into understanding the catalytic mechanism of MBOATs. However, to make the story more convincing, some concerns and problems need to be solved first by the authors. My points are as follow:

(1) The topology proposed in this manuscript against all previous proposed topology for other MBOATs (DltB, ACAT1 or SOAT1, DGAT, and HHAT). The authors need to provide more experimental evidence to prove that their proposed topology is correct. Here I want to remind the authors of two facts: (i) for multi-pass transmembrane proteins without signal peptide, it's not that rare to have their N terminal in the luminal/or extracellular side, for example DltB which also belongs to MBOAT superfamily, has both of its N-terminus and C-terminus on the extracellular side. (ii) The donor substrates of acyl-CoA originally exist in the cytosolic side which make a reversed topology like other MBOATs more reasonable.

(2) In the density map for cLPCAT3/LPC, the density for LPC head group is clear, however there are two continuous densities near to the connection between LPC head group and hydrophobic tail, pointing upwards and downwards respectively. The authors built the LPC tail into the density pointing upwards and left the density pointing downwards, which seems to be visually more continuous with the head group, unmodeled (Fig. 4D). However, in the map of cLPCAT3/araCoA at similar position compared to the density pointing upwards mentioned above, there is also a similar uncharacterized density. I'm wondering whether the authors have built the LPC tail into density belongs to some other molecule. If the tail is built into the density pointing downwards, the structural findings will be in consistence with the lateral gates observed in ACAT1 and DGAT1 which have been speculated to be the entrance of acyl acceptor. The authors need to make a more careful tracing, comparison and proper discussion.

(3) For purified cLPCAT3 mutants, gel filtration profiles should be shown, especially those significantly decreased the activity in Fig. 2e and Fig. 3d, in order to rule out the possibility that mutations affect protein folding.

(4) In Fig. 6D, the intensity of monomer bands for wild type is much weaker than the two mutants. It's better to make the intensity of those wild type bands comparable to that of the mutants.

(5) Line 234, "thr" may should be "the".

Reviewer #2 (Remarks to the Author):

LPCAT3 is a MBOAT family protein that catalyze the acylation of lysophosphatidylcholine. Despite its importance, the structure and mechanism of LPCAT3 remain elusive. Here, Zhang et al. present three structures of chicken LPCAT3, in apo, acyl-CoA bound and LPC bound state. These structures are novel and represent a major breakthrough for understanding the mechanism of LPCAT3 proteins. They also shed light on the mechanism of MBOAT family proteins. The structural work is technically solid. I believe this manuscript will surely be the first of many papers that will eventually reveal the mechanisms of LPACT3. I have one major issue and a few minor issues that might be helpful for the authors to improve the manuscript:

Major issue:

The author proposed the membrane topology of LPCAT3 is in opposite to other MOBAT family members, because there is no N-terminal signal peptide. More importantly, this topology model

proposes that the acyl-CoA accesses LPCAT3 from ER lumen. This is in great contrast to my current knowledge that acyl-CoA mainly distributes in cytosol. On the other hand, there are a handful membrane proteins, of which N-terminus are outside but without signal peptide. Therefore, I would suggest the authors to perform experiments to confirm the current proposed topology or carefully revise the topology model by citing literatures.

Minor issues:

1. The angular distribution should be presented in supplementary figures.
2. The coloring scheme of local resolution is different from canonical rainbow colors. I would suggest to change it to the common one.
3. Both X-ray table S1 and cryo-EM table seems incomplete and contains errors. Eg. Do the last two rows for cryo-EM table represent Ramachandran distributions? Are the numbers inverted? The tables need to be carefully revised.
4. Is it possible to perform LC-MS experiments to identify the ligands observed in line 219-220?

We thank reviewers for their valuable comments on our manuscript. Based on those feedbacks, we have made revisions to our previous draft accordingly. The reviewer's comments are laid out below with gray background and our response is given in normal font and changes/additions to the manuscript are highlighted in yellow.

Reviewer #1 (Remarks to the Author):

Qing Zhang et al solved the crystal structure of apo-LPCAT3 and cryo-EM structures of acyl-donor and acyl-acceptor bounded states of LPCAT3. It is the first time for an acyl-acceptor substrate to be experimentally visualized in 3D structure for MBOATs that take lipids as acyl group acceptors in acyl transfer reaction. The results in this manuscript will provide important insights into understanding the catalytic mechanism of MBOATs. However, to make the story more convincing, some concerns and problems need to be solved first by the authors. My points are as follow:

(1) The topology proposed in this manuscript against all previous proposed topology for other MBOATs (DltB, ACAT1 or SOAT1, DGAT, and HHAT). The authors need to provide more experimental evidence to prove that their proposed topology is correct. Here I want to remind the authors of two facts: (i) for multi-pass transmembrane proteins without signal peptide, it's not that rare to have their N terminal in the luminal/or extracellular side, for example DltB which also belongs to MBOAT superfamily, has both of its N-terminus and C-terminus on the extracellular side. (ii) The donor substrates of acyl-CoA originally exist in the cytosolic side which make a reversed topology like other MBOATs more reasonable.

Our reply: We understand the concern from both reviewers about the membrane topology of LPCAT3 because this also puzzled us when solving the structures. We believe this reverse topology of LPCAT3 is a result of its transmembrane helices composition: the LPCAT3 lack one transmembrane helix at its N-terminal compared with SOAT1 and DGAT1¹⁻⁴, which makes LPCAT3 a reverse orientation against that of SOAT and DGAT1. Just as pointed out by both reviewers, our model relies on the presumption that all three MBOATs have their N-terminal localized at the cytoplasmic side of the membrane since none of them has a signaling peptide and some membrane proteins without signaling peptide, such as bacterial MBOAT DltB, might have their N-terminal in the luminal/or extracellular side⁵. Therefore, it is critical to know if LPCAT3 has its N-terminal localize in ER lumen/extracellular side or not. To further study the actual localization of the N/C-terminus of LPCAT3, we conducted the protease digestion experiment on the HEK293 cells overexpressing the LPCAT3. Since part of ER-localized membrane could be found on cell membrane when overexpressed, the exposure of their N/C-terminus to the extracellular environment could be

detected by applying sequence-specific proteases (TEV and HRV-3C proteases in this assay) to intact cells overexpressing LPCAT3 with the recognition sequences for protease at N-terminal or C-terminal. In our experiments (shown in Supplementary Figure 10), the western blotting showed that only the C-terminal of LPCAT3 is accessible to the protease attack and the N-terminal of LPCAT3 keeps uncleavable upon the protease treatment. To exclude the possibility of low proteolytic efficiency on LPCAT3 in a membrane-bound state, we disrupted the cells to conduct the same assay on membrane preparations and found that both N- and C-terminal of LPCAT3 are now cleavable upon the protease treatment. Based on these new data, we believe our current molecular model of cLPCAT3 might reflect its natural topology on the ER membrane. We agree that the opposing topology proposed by the reviewers aligns very well with the cytoplasmic distribution of cellular araCoA molecules and makes the entry of araCoA into the reaction chamber more straightforward. However, both the structural analysis and biochemical assay in our studies consistently support an ER lumen-facing acyl-CoA entrance, and thus we keep the original topology shown in Figure 1. To properly reflect the concerns from reviewers, we made the further discussion to detail the evidence supporting our model and show the readers the difference between LPCAT3 and other MBOATs.

(2) In the density map for cLPCAT3/LPC, the density for LPC head group is clear, however there are two continuous densities near to the connection between LPC head group and hydrophobic tail, pointing upwards and downwards respectively. The authors built the LPC tail into the density pointing upwards and left the density pointing downwards, which seems to be visually more continuous with the head group, unmodeled (Fig. 4D). However, in the map of cLPCAT3/araCoA at similar position compared to the density pointing upwards mentioned above, there is also a similar uncharacterized density. I'm wondering whether the authors have built the LPC tail into density belongs to some other molecule. If the tail is built into the density pointing downwards, the structural findings will be in consistence with the lateral gates observed in ACAT1 and DGAT1 which have been speculated to be the entrance of acyl acceptor. The authors need to make a more careful tracing, comparison and proper discussion.

Our reply: We thank the reviewer for pointing out the potential issue in assigning the sn-1 acyl chain in the non-protein electron density of the cLPCAT3/LPC EM map. First of all, the more continuity of unmodeled density (Figure 4d in both original and revised manuscript) is not sufficiently long for fitting a 12:0-acyl carbon chain since the end-piece (about 1/3 length) of the blob of electron density is too close to the protein backbone. In structure solving, we used the LigandFit module of the Phenix program package for the automatic ligand fitting inside the reaction chamber, and only the electron density within the horizontal tunnel was identified as and fit with the LPC

molecule. Upon the reviewer's suggestion, we manually fit the LPC (12:0) into the unmodeled density as shown in supplementary figure 9A and in those models, neither of the acyl chain in two potential conformations (shown as a yellow and a cyan stick model, respectively) could match the density very well. In addition, this unmodeled density is identified in the "side pocket" of LPCAT3, instead of the downwards vertical tunnel which is in general empty in the map of cLPCAT3/LPC. The "side pocket" of LPCAT3 is a small cavity with only one open near to the joint point for the horizontal-vertical tunnels, and thus this reaction chamber and non-protein electron density might not support the entrance function for the lateral gate of LPCAT3 as well as that of ACAT1/DGAT1. On the other hand, we also carefully check the uncharacterized density in the map of cLPCAT3/araCoA and found that this density is of an even poorer quality to assign with any substrate molecules reliably. Therefore we left it unmodeled and hopefully in future studies an enzyme-product structure could clarify its identity.

Based on the EM map, the molecular models for the LPCAT3-ligand complex, and the manually tracing on the non-protein electron density inspired by the reviewer's comment, we believe the original ligand-binding models in our manuscript might be the best solution for the experimental data. However, we appreciate the reviewer for bringing to our notice that the density shown in figure 4D could be confusing to the readers. We have made revisions and corrections accordingly as follows:

- 1) We re-created Figure 4D to take the steric hindrance into account when generating the surface model for the unmodeled electron density.
- 2) To let the readers understand another possibility exists for the LPCAT3-LPC interaction and the reason we consider this possibility might not be the truth, we showed the manually modeled LPC molecules in alternative conformations in supplementary Figure 9A and compared those different models in the discussion part of the main text.
- 3) We also show the uncharacterized density in LPCAT3-araCoA in Figure 9B and made discussion correspondingly.
- 4) We attached the coordinate files with the LPC molecules in two manually modeled conformations for further examination by reviewers.

(3) For purified cLPCAT3 mutants, gel filtration profiles should be shown, especially those significantly decreased the activity in Fig. 2e and Fig. 3d, in order to rule out the possibility that mutations affect protein folding.

Our reply: It is a nice suggestion and we agree that further evidence should be provided to confirm that the change in enzymatic activities is not a result of global folding defects of cLPCAT3 mutants. We attached the GF profiles for the wild type and all the cLPCAT3 mutants in supplementary Figure 11. The GF profiles of mutants showed that all the mutants in functional assay maintain proper folding. Even the mutant with the lowest stability, cLPCAT3 K297A, has a

large proportion of proteins in proper folding, allowing a reliable measurement of the effect of the mutations on enzymatic activities.

(4) In Fig. 6D, the intensity of monomer bands for wild type is much weaker than the two mutants. It's better to make the intensity of those wild type bands comparable to that of the mutants.

Our reply: We thank the reviewer for pointing out the data quality issue in the wild-type control of the crosslinking assay. A further WB assay has been conducted using proper amounts of wild type cLPCAT3 in the crosslinking assay and Figure 6d was updated with the blotting bands for wild type control in a comparable intensity to that of the mutants.

(5) Line 234, “thr” may should be “the”.

Our reply: Thanks, and we have corrected the typo according to the comment.

Reviewer #2 (Remarks to the Author):

LPCAT3 is a MBOAT family protein that catalyze the acylation of lysophosphatidylcholine. Despite its importance, the structure and mechanism of LPCAT3 remain elusive. Here, Zhang et al. present three structures of chicken LPCAT3, in apo, acyl-CoA bound and LPC bound state. These structures are novel and represent a major breakthrough for understanding the mechanism of LPCAT3 proteins. They also shed light on the mechanism of MBOAT family proteins. The structural work is technically solid. I believe this manuscript will surely be the first of many papers that will eventually reveal the mechanisms of LPACT3. I have one major issue and a few minor issues that might be helpful for the authors to improve the manuscript:

Major issue:

The author proposed the membrane topology of LPCAT3 is in opposite to other MOBAT family members, because there is no N-terminal signal peptide. More importantly, this topology model proposes that the acyl-CoA accesses LPCAT3 from ER lumen. This is in great contrast to my current knowledge that acyl-CoA mainly distributes in cytosol. On the other hand, there are a handful membrane proteins, of which N-terminus are outside but without signal peptide. Therefore, I would suggest the authors to perform experiments to confirm the current proposed topology or carefully revise the topology model by citing literatures.

Our reply: Just as our response to the comment (1) from the other reviewer, we understand this concern about the topology of LPCAT3 and agree with the reviewer that, everything could be explained more easily if the LPCAT3 adapts to a “classic” conformation with its entry tunnel for araCoA facing to the cytoplasmic side of the ER membrane. As discussed in our response to reviewer

#1, to find out the actual localization of the N/C-terminus of LPCAT3, we conducted a protease digestion experiment where the cell overexpressing cLPCAT3 with flag/strep tag at its N- or C-terminal cleavable by the sequence-specific protease such as TEV or HRV-3C protease. The result showed that the N-terminal tag of LPCAT3 within intact cells cannot be cut by the protease treatment, while the C-terminal tag could be efficiently cut, indicating that the topology of LPCAT3 proposed by our structure could be correct and the N- and C-terminal of LPCAT3 might be localized in cytoplasmic and lumen/extracellular side, respectively. As a control, the cells overexpressing LPCAT3 were disrupted into membrane pieces, and then both the N- and C-terminal tags could be cut by the corresponding proteases, confirming the high cleavage efficiency of the proteases on the membrane-bound LPCAT3. Therefore, the topology currently proposed matches the biochemical results very well and thus we kept the original molecular model in the revised manuscript. In addition, we made further discussion in the main text about the concerns of topology and showed the results of the protease digestion assay on intact cells in supplementary figure 10.

Minor issues:

1. The angular distribution should be presented in supplementary figures.

Our reply: We thank the reviewer for the suggestion and the parameters have been added into Figure 5e and 6e.

2. The coloring scheme of local resolution is different from canonical rainbow colors. I would suggest to change it to the common one.

Our reply: The coloring scheme of the local resolution has been changed to the classic one in supplementary Figure 5f and 6f.

3. Both X-ray table S1 and cryo-EM table seems incomplete and contains errors.

Eg. Do the last two rows for cryo-EM table represent Ramachandran distributions? Are the numbers inverted? The tables need to be carefully revised.

Our reply: We thank the reviewer for pointing out the errors in the supplementary tables and we have double-checked the whole table to correct the issues. In addition, more parameters were provided in tables to allow a comprehensive evaluation and review of the data quality of crystallography and cryogenic electron microscopy research.

4. Is it possible to perform LC-MS experiments to identify the ligands observed in line 219-220?

Our reply: We understand the concern from the reviewer. LPCAT3 serves as a polyunsaturated acyl transferase in phosphatidylcholine remodeling, however, its catalytic selectivity on the acyl chain to be transferred and acceptors, i.e., LPCs, are not highly strict. Previous biochemical studies showed that a group of acyl-CoA molecules with varied lengths and number of double-bonds,

such as arachidonoyl-CoA (20:4), linolenoyl-CoA (18:3), linoleoyl-CoA (18:2), and a group of LPCs with varied lengths, such as 10:0, 12:0, 14:0, 16:0, 18:0, could be utilized as the substrates for LPCAT3 mediated reaction.⁶⁻⁸. In our cryo-EM studies on LPCAT3-LPC, we only added 12:0-LPC as the acyl acceptor and no exogenous acyl-CoA was supplemented. Therefore, the unmodeled electron density, mentioned in line 219-220, represented the acyl chain on the sn-2 position of PCs which are co-purified with LPCAT3. Therefore, the PC products should contain various acyl chains derived from a pool of endogenous polyunsaturated fatty acids. Thus, it is difficult to detect their identities, specifically derived from line 219-220, by LC-MS methods. We agree with the reviewer that identifies the density of product molecules could help us to understand the enzymatic mechanism and selectivity of LPCAT3. The future efforts should be made to determine the structure of LPCAT3 in complex with PC products to elucidate more details in acyltransferring reaction.

Reference

- 1 Guan, C. *et al.* Structural insights into the inhibition mechanism of human sterol O-acyltransferase 1 by a competitive inhibitor. *Nat Commun* **11**, 2478, doi:10.1038/s41467-020-16288-4 (2020).
- 2 Qian, H. *et al.* Structural basis for catalysis and substrate specificity of human ACAT1. *Nature* **581**, 333-338, doi:10.1038/s41586-020-2290-0 (2020).
- 3 Sui, X. *et al.* Structure and catalytic mechanism of a human triacylglycerol-synthesis enzyme. *Nature* **581**, 323-328, doi:10.1038/s41586-020-2289-6 (2020).
- 4 Wang, L. *et al.* Structure and mechanism of human diacylglycerol O-acyltransferase 1. *Nature* **581**, 329-332, doi:10.1038/s41586-020-2280-2 (2020).
- 5 Ma, D. *et al.* Crystal structure of a membrane-bound O-acyltransferase. *Nature* **562**, 286-290, doi:10.1038/s41586-018-0568-2 (2018).
- 6 Zhao, Y. *et al.* Identification and characterization of a major liver lysophosphatidylcholine acyltransferase. *J Biol Chem* **283**, 8258-8265, doi:10.1074/jbc.M710422200 (2008).
- 7 Hishikawa, D. *et al.* Discovery of a lysophospholipid acyltransferase family essential for membrane asymmetry and diversity. *Proc Natl Acad Sci U S A* **105**, 2830-2835, doi:10.1073/pnas.0712245105 (2008).
- 8 Jain, S. *et al.* Characterization of human lysophospholipid acyltransferase 3. *J Lipid Res* **50**, 1563-1570, doi:10.1194/jlr.M800398-JLR200 (2009).

REVIEWER COMMENTS

Reviewer #1 (Remarks to the Author):

The authors have done substantial work during the revision to address my concerns raised on the initial manuscript. The authors provide experimental evidence that is in consistence with their proposed topology model, although the topology is against previous MBOATs structure models. The biochemical work shown in the revised manuscript is well controlled; careful density tracing and proper discussion convincingly support the correctness of the structure. Overall, the entire manuscript has been improved significantly after the revision. I support the publication of this work at Nature Communications.

Reviewer #2 (Remarks to the Author):

The author have made substantial revisions to the manuscript and answered all of my questions. But I am still conservative about the topology issue:

In the revision, they authors have provided more data to support the topology assignment proposed in the manuscript. Because this issue is highly important to the MBOAT field, I would suggest more rigorous validation. The following suggestions on detailed experimental procedures might be helpful.

1. Did the authors add linker between protease cutting site and LPCAT3? I would suggest add flexible linker like GSGSGS between them to improve the cleavage efficient on intact membrane.
2. Is it possible to use both N terminal flag tag and C terminal strep tag, and with HRV 3C protease cleavage sequence for both tags? This design will eliminate the difference in protease efficiency.
3. I would suggest to use irreversible protease inhibitor such as iodoacetamide to stop the cleavage reaction instead of washing them out, to avoid residual protease contamination.
4. The authors need to show the constructs used in the protease cleavage experiment are functional.
5. The authors need to state how many times the experiments were repeated successfully in the figure legend.
6. The data in Fig. S11 suggests a large fraction of over expressed LPCAT3 is on the plasma membrane. If the topology proposed by the author is correct, there should be a high LPCAT3 activity in live cells when supplementing LC-CoA substrate to the extracellular solution, but control cells have not. Is it possible to detect such activity?

We thank reviewers for their valuable comments on our manuscript. Based on those feedbacks, we have made revisions to our previous draft accordingly. The reviewer's comments are laid out below with gray background and our response is given in normal font and changes/additions to the manuscript are highlighted in yellow.

Reviewer #1 (Remarks to the Author):

The authors have done substantial work during the revision to address my concerns raised on the initial manuscript. The authors provide experimental evidence that is in consistence with their proposed topology model, although the topology is against previous MBOATs structure models. The biochemical work shown in the revised manuscript is well controlled; careful density tracing and proper discussion convincingly support the correctness of the structure. Overall, the entire manuscript has been improved significantly after the revision. I support the publication of this work at Nature Communications.

Reviewer #2 (Remarks to the Author):

The author have made substantial revisions to the manuscript and answered all of my questions. But I am still conservative about the topology issue: In the revision, they authors have provided more data to support the topology assignment proposed in the manuscript. Because this issue is highly important to the MBOAT field, I would suggest more rigorous validation. The following suggestions on detailed experimental procedures might be helpful.

1. Did the authors add linker between protease cutting site and LPCAT3? I would suggest add flexible linker like GSGSGS between them to improve the cleavage efficient on intact membrane.

Our reply: We agreed with the reviewer about the concerns that the different working efficiency of protease on N-/C-terminal tag could also produce a similar result as shown in Supplementary Fig. 10b. Therefore, in this further revised manuscript, we added GS linker between the LPCAT3 and both N- and C-terminal tags as suggested by the reviewer to minimize the difference on the cleavage efficiencies.

2. Is it possible to use both N terminal flag tag and C terminal strep tag, and with HRV 3C protease cleavage sequence for both tags? This design will eliminate the difference in protease efficiency.

Our reply: We believe this is a considerate suggestion which could improve the

reliability of the protease cleavage experiment. We thus re-designed the expression construct which contains a flag tag at the N-terminal and a strep tag at the C-terminal of LPCAT3, both tags spaced by a HRV 3C recognition sequence and GS-linker sequences. A schematic diagram of the expression construct was shown in Supplementary Fig. 10b. In this assay using new constructs, similar results were observed and when treated with HRV-3C protease, the C-terminal strep tag was efficiently removed as indicated by the reduced intensity of blotting band, while the N-terminal tag was in general keep its band intensity unchanged with a slight shift observed due to the loss of C-terminal tag.

3. I would suggest to use irreversible protease inhibitor such as iodoacetamide to stop the cleavage reaction instead of washing them out, to avoid residual protease contamination.

Our reply: Thanks for the suggestion and we added the iodoacetamide at a concentration of 0.1 mg mL^{-1} to stop the cleavage reaction, as detailed in supplementary method.

4. The authors need to show the constructs used in the protease cleavage experiment are functional.

Our reply: In this further revised manuscript, a functional assay has been conducted and the results showed that this new construct generated proteins with normal acyltransferase activities.

5. The authors need to state how many times the experiments were repeated successfully in the figure legend.

Our reply: all the analytic experiments in our manuscript have been conducted in triplicate and we have added this information in the figure legends in the further revised manuscript.

6. The data in Fig. S11 suggests a large fraction of over expressed LPCAT3 is on the plasma membrane. If the topology proposed by the author is correct, there should be a high LPCAT3 activity in live cells when supplementing LC-CoA substrate to the extracellular solution, but control cells have not. Is it possible to detect such activity?

Our reply: Thanks for the suggestion. Although on-cell enzymatic assay were previously used in detecting the expression level of membrane-bound enzyme and proved quite efficient in screen expression constructs, this method could usually provide the information about express level of specific enzyme and it will be difficult to probe the orientation of the reaction pocket or the

substrate accessibility, since the substrates we supplied, such as long chain acyl-CoA and Lysolipids, are able to be incorporated into membrane and enter the cytosolic side. This kind of enzymatic assay was usually semiquantitative and its accuracy was limited, which might not provide reliable data. In addition, it will be difficult to conduct a control assay to set a benchmark for the standard activities of inward- and outward-facing enzyme. Therefore, we believe our protease digestion assay might be a suitable choice for the functional analysis in our structural report.

We understand the concerns from the reviewer since this unexpected topology, if wrongly proposed, might cause confusion in readers. In manuscript, we added a statement to express the conflict between the topology LPCAT3 and previously reported MBOATs, as well as the principles and limitations of our protease digestion assay. We thank the reviewer for the suggestions and we found them have helped us to significantly improve the integrity and reliability of our manuscript. We hope those revisions, along with the newly designed and conducted assay, would make our manuscript more suitable for publication in Nature Communication.

REVIEWER COMMENTS

Reviewer #2 (Remarks to the Author):

The authors have provided convincing evidence for the new topology of LPCAT3. I have no more questions.